# OpenReview forum: "Automatic Robustness Stress Testing of LLMs  as Mathematical Problem Solvers"
_ICLR.cc/2026/Conference — ICLR 2026 Conference Withdrawn Submission_

### Official Review · Reviewer_eNkS · 2025-10-24

**Soundness:** 2
**Presentation:** 3
**Contribution:** 2
**Rating:** 4
**Confidence:** 3

**Summary:**

This paper introduces AR-CHECKER, a framework to generate mathematical problem variants that maintain the semantic meanings of the original one but might fail the LLMs in solving it. The approach uses 3 LLM components: a rewriter, verifier, and target LLM. The framework uses multi-round parallel streams as well to explore the space of challenging variants. Empirically, GSM8K and MATH-500 demonstrate significant drops in performance when applying AR-CHECKER.

**Strengths:**

The paper introduces a novel framework for modifying traditional stress testing methods. The paper designed a novel method in changing the semantic meanings of the original one and shows that models fail after modifying the semantic meaning. The analysis is thorough, backing up the method. In addition, the method shows that accuracy decreases abruptly when applying AR-CHECKER, and is generalizable towards other domains beyond mathematical reasoning. This could potentially show data contamination and insufficient stress testing in benchmarks.

**Weaknesses:**

1. I am slightly unconvinced with the manual annotation in table 3. While table 3 shows 98% precision, this is based on only 150 samples. More qualitative analysis should be done towards the accuracy, as the model is well known to generate errors in rephrasing. It would be great to discuss the F1 and recall with more detailed analysis (and with samples beyond 150). Given that the method is using GSM8K, it would be great to use verifiers such as Python to detect the correctness of the entire samples.

2. How would this differ from GSM-Symbolic, where their core contribution is to convert GSM8K by changing the variables, rephrasing, and others? I don't quite understand how GSM-Symbolic's methodology showing contamination within GSM8K relates to how applying AR-CHECKER into GSM-Symbolic helps.

3. What are the failure examples of the model generating variants? For example, when does AR-CHECKER fail to find challenging variants? The paper shows some models have lower TFR (e.g., 33.33% for Qwen2.5-7B-Instruct on GSM8K) but doesn't deeply analyze why.

4. The paper reports point estimates without error bars or measures of variance, with temperature=1. Statistical analysis should be provided in this case.

5. Multiple-choice questions such as those in MMLU should show the examples further. Do the answer choices change as well or just the question?

**Questions:**

1. Could you provide more information on the false negative rate of the verifier? What happens when the verifier incorrectly judges that meaning has changed?

2. While multiple-choice options are applied, how does the robustness vary based on [1]?

3. It would be great to clarify the core contribution of GSM-Symbolic (since they focused on changing the questions as well).

4. Would it be possible to verify more samples for accuracy compared to the 100/150 samples of manual annotation?

5. Are there examples showing how the rewrites evolve during the multi-round iteration process?

6. AR-CHECKER is only applied to traditional question answering for evaluation, such as CoT. Does method such as Faithful CoT [2], CoMAT [3], or ToT [4] others mitigate the decrease in errors?

[1] Gupta, Vipul, et al. "Changing answer order can decrease mmlu accuracy." arXiv preprint arXiv:2406.19470 (2024).

[2] Lyu, Qing, et al. "Faithful chain-of-thought reasoning." The 13th International Joint Conference on Natural Language Processing and the 3rd Conference of the Asia-Pacific Chapter of the Association for Computational Linguistics (IJCNLP-AACL 2023). 2023.

[3] Leang, Joshua Ong Jun, Aryo Pradipta Gema, and Shay B. Cohen. "Comat: Chain of mathematically annotated thought improves mathematical reasoning." arXiv preprint arXiv:2410.10336 (2024).

[4] Yao, Shunyu, et al. "Tree of thoughts: Deliberate problem solving with large language models." Advances in neural information processing systems 36 (2023): 11809-11822.

---

> ### Author Response · Authors · 2025-11-25
>
> Thanks for your thoughtful and constructive feedback.
>
> ---
>
> ### **Q1. Clarity of Contribution**
>
> **R1.**
> Our work does *not* aim to restate conclusions from GSM-Symbolic or similar studies showing that perturbing questions can reveal LLM unreliability. Instead, our core contribution is a **dynamic and model-adaptive evaluation framework**.
> Unlike static datasets—such as GSM-Symbolic—that rely on fixed templates or rules and therefore suffer from inherent risks of data contamination, our framework:
>
> - can be **applied dynamically to each target model**,
> - scales naturally as LLMs improve,
> - allows the use of **stronger rewriters to evaluate stronger target models**.
>
> This represents a fundamental distinction from static benchmark construction.
>
> ---
> ### **Q2. In-depth Analysis of MMLU**
>
> **R2.**
> Thank you for these insightful questions. Regarding Q5, we clarify that AR-CHECKER primarily attacks the question stem (the problem description) to introduce semantic variations—such as increasing linguistic complexity or adding irrelevant context—while keeping the answer choices and their order unchanged.
>
> This design choice allows us to isolate semantic robustness from the positional bias discussed in [1]. While we acknowledge that changing answer order can degrade accuracy, our experiments maintain the original option arrangement. Consequently, the observed performance drops are driven by the model's failure to process the semantically rewritten context rather than sensitivity to answer placement. The table below demonstrates a specific example where the question is reformulated while the options remain fixed:
>
> |      | Vanilla | AR-CHECKER |
> | :--- | :---: | :---: |
> | Question | The following are multiple choice questions (with answers) about high school computer science. Let x = 1. What is x << 3 in Python 3?\nA. 1\nB. 3\nC. 8\nD. 16\nAnswer: | The following are multiple choice questions (with answers) about high school computer science. Given an integer variable x initialized to 1, what is the result of applying a bitwise left shift operation with a shift count of 3 in Python 3? Note that the bitwise left shift operation (<<) shifts the bits of the number to the left by the specified count, effectively multiplying the number by 2 raised to the power of the shift count.\nA. 1\nB. 3\nC. 8\nD. 16\nAnswer: |
> | Answer | To solve this question, we need to understand the concept of left shift operator in Python.\n\nThe left shift operator in Python is `<<`. It shifts the bits of the number to the left by a specified number of places.\n\nIn this case, `x = 1` and `x << 3` means we are shifting the bits of `x` to the left by 3 places.\n\nSince `x` is a single digit number (1), shifting it to the left by 3 places will result in a number with 3 zeros after the decimal point.\n\nSo, `x << 3` is equivalent to `1 << 3`.\n\nUsing the Python `math` module, we can calculate the result of `1 << 3` as follows:\n\n```python\nimport math\n\nx = 1\nresult = x << 3\nprint(result)  # Output: 8\n```\n\nTherefore, the correct answer is C. 8. | The correct answer is D. 16.\nThe bitwise left shift operation (<<) shifts the bits of the number to the left by the specified count, effectively multiplying the number by 2 raised to the power of the shift count. In this case, the shift count is 3, so the bits of the number 1 are shifted 3 positions to the left.\nThe binary representation of 1 is 00000001. Shifting it 3 positions to the left results in 10000000, which is equivalent to the decimal value 16. |
>
>
> After the perturbation by AR-CHECKER, the question became more semantically complex and descriptive. Additional phrases, such as "Note that," disrupted the model’s attention, causing it to focus on secondary information and overlook the core of the problem. Consequently, the value 00000001 was incorrectly left-shifted by 7 bits instead of 3 bits. Furthermore, the model also miscalculated 10000000 (decimal 128) as decimal 16.
>
> ---
>
> ### **Q3. Expanded Human Verification and Discussion on Metrics **
>
> **R3.**
> We appreciate your suggestion to expand the validation scope. To address the concern regarding sample size, we conducted an additional round of human annotation on 120 new samples (60 from GSM8K and 60 from MATH-500) that were classified as valid adversarial examples by our judge model. The human verification results showed an accuracy of 59/60 for GSM8K and 58/60 for MATH-500, yielding a consistent precision of ~97-98%. This strongly validates the scalability and reliability of our automated verifier beyond the initial small subset.

---

> > ### Author Response · Authors · 2025-11-25
> >
> > ### **Q4. Why not apply advanced mathematical solving pipelines such as Faithful CoT, CoMAT， or ToT ?**
> >
> > **R4.**
> > Our study focuses on evaluating the **standard generation-based usage mode**, which is the most widely used in practice. The robustness of this mode is directly important for real-world deployment, and its performance generally correlates with trends observed in more advanced mathematical-reasoning pipelines.
> > These specialized methods fall outside the main scope of our work, which aims to assess robustness under typical generation settings rather than specialized solving frameworks.

---

> ### Author Response · Authors · 2025-11-25
>
> ### **Q5. Evolution of Rewrites via Multi-Round Iteration**
>
> **R5.**
>
> The table below demonstrates a representative example of how the rewritten queries evolve during the multi-round iteration process to generate a successful adversarial variant:
>
> The initial question is ```Marie ordered one chicken meal that costs $12, 5 packs of milk that costs $3 each, 4 apples that cost $1.50 each, and some boxes of pizza. Marie paid a total of $50. How many boxes of pizza did Marie order if each box costs $8.50?```
>
> | Iteration | Question | Response | Analysis |
> |---|---|---|---|
> | 1 | Marie went to a local grocery store where she purchased 4 apples at `$`1.50 each, some boxes of pizza, a chicken meal priced at `$`12, and 5 packs of milk, each costing `$`3. Her total bill came to `$`50. How many boxes of pizza, priced at `$`8.50 each, did Marie order? | To solve this problem, let's break it down step by step.\n\nFirst, let's calculate the total cost of the items Marie bought without the pizza boxes:\n\n- 4 apples at `$`1.50 each: 4 * `$`1.50 = `$`6\n- 1 chicken meal at `$`12: `$`12\n- 5 packs of milk at `$`3 each: 5 * `$`3 = `$`15\n\n...So, Marie ordered 2 boxes of pizza. | In the first iteration, the rewriter changed the order of the items and the way the question was phrased, but did not introduce any additional information. The target model answered the question correctly, and the judge model classified this instance as C. |
> | 2 | Marie ordered one chicken meal that costs `$`12, 5 packs of milk that costs `$`3 each, 4 apples that cost `$`1.50 each, 2 bottles of water that cost `$`0.50 each, and some boxes of pizza. Additionally, she received a discount of `$`2 on her order. Marie paid a total of `$`50. How many boxes of pizza did Marie order if each box costs `$`8.50? | Hello Marie, I'm here to help you with your order. Let's break down your purchase and calculate the total amount you paid.\n\nYou ordered one chicken meal for `$`12. \n5 packs of milk that cost `$`3 each is 5 * `$`3 = `$`15.\n4 apples that cost `$`1.50 each is 4 * `$`1.50 = `$`6.\n2 bottles of water that cost `$`0.50 each is 2 * `$`0.50 = `$`1.\nTotal cost of non-pizza items is `$`12 + `$`15 + `$`6 + `$`1 = `$`34.\n\nYou also received a discount of `$`2 on your order, so the total amount you paid is `$`50 - `$`2 = `$`48.\n\nNow, let's calculate the total cost of pizza. The total amount you paid is `$`48, and the cost of non-pizza items is `$`34. To find the cost of pizza, subtract the cost of non-pizza items from the total amount paid:\n`$`48 - `$`34 = `$`14.\n\nSince each box of pizza costs `$`8.50, divide the total cost of pizza by the cost per box to find the number of boxes:\n`$`14 \/ `$`8.50 = 1.64\n\nTo find the whole number of boxes, round down to 1, as you can't order a fraction of a box. Therefore, Marie ordered 1 box of pizza. | In the second iteration, the rewriter additionally added two bottles of water priced at `$`0.50 each and a `$`2 discount. The model attempt to use the discount to offset the added cost of the water, but it made a calculation error that ultimately altered the semantics of the problem. Moreover, in its response, the model incorrectly subtracted the discount from the payment amount instead of adding it back, resulting in an erroneous computation and a deviation from the original answer. The judge model classified this instance as A. |
> | 3 | Marie ordered one chicken meal for `$`12, 5 packs of milk at `$`3 each, and 4 apples at `$`1.50 each. She initially picked up 3 bottles of water for `$`0.75 each, but realized her mistake and returned them before checkout. Marie paid a total of `$`50. Ignoring the water bottles, how many boxes of pizza costing `$`8.50 each did Marie order? | To find out how many boxes of pizza Marie ordered, we need to break down her order and subtract the water bottles.\n\nMarie ordered:\n1. Chicken meal: `$`12\n2. 5 packs of milk: 5 * `$`3 = `$`15\n3. 4 apples: 4 * `$`1.50 = `$`6\n4. 3 bottles of water: 3 * `$`0.75 = `$`2.25\n\nTotal cost without water bottles: `$`12 + `$`15 + `$`6 + `$`2.25 = `$`35.25\n\nSince Marie paid a total of `$`50, we need to subtract the cost of the water bottles to find the cost of the pizza:\n\n`$`50 - `$`2.25 = `$`47.75\n\nNow, we need to find out how many boxes of pizza Marie ordered. We know the cost of one box of pizza is `$`8.50.\n\nNumber of boxes = Total cost \/ Cost per box\n= `$`47.75 \/ `$`8.50\n= 5.5\n\nSince we can't order a fraction of a box, we need to round down to the nearest whole number. Therefore, Marie ordered 5 boxes of pizza. | In the third iteration, the rewriter correctly added extra information that did not affect the original semantics. However, the target model failed to recognize that the additional information “bottled water” was irrelevant to solving the problem and incorrectly included its cost in the total price of the items, resulting in an incorrect answer. The judge classified this instance as D, and the iteration ended. |

---

> > ### Comment · Reviewer_eNkS · 2025-11-27
> >
> > I thank the reviewer for their clarifications. However, I will be keeping my score as it is.
> >
> > 1. The results raise concerns about performance on GSM-Symbolic and other domains. The main issue is not merely analysing perturbations, but analysing what happens if the observed behaviours change due to contamination or genuine generalisability in a given domain. Since GSM-Symbolic evaluation follows the same structure as GSM8K, it would be helpful to examine whether the perturbations meaningfully affect those.
> >
> > 2. It would be valuable to report performance on a subset of MMLU [1] or MMLU-Redux [2], given time constraints. This would help clarify whether perturbations hinder the generalizability of AR-CHECKER across broader domains.
> >
> > 3. I respectfully disagree with the argument provided. A baseline comparison is still necessary to determine whether overall performance improves or degrades. The absence of comparisons with existing methods would be concerning, as it limits the ability to contextualize the effectiveness of the proposed approach.
> >
> > [1] Hendrycks, Dan, et al. "Measuring massive multitask language understanding." arXiv preprint arXiv:2009.03300 (2020).
> >
> > [2] Gema, Aryo Pradipta, et al. "Are we done with mmlu?." Proceedings of the 2025 Conference of the Nations of the Americas Chapter of the Association for Computational Linguistics: Human Language Technologies (Volume 1: Long Papers). 2025.

---

### Official Review · Reviewer_jDpR · 2025-10-31

**Soundness:** 4
**Presentation:** 3
**Contribution:** 3
**Rating:** 6
**Confidence:** 3

**Summary:**

This paper propose to evalaute the LLM robustness regarding mathmatical problems. To implement the evaluation, the authors design a framework consists of a rewriter (to paraphrase the math problem), a verifier (to checker wheter the initial core meaning is preserved), and a target LLM. Extensive experiments on various benchmarks reveal the vulnerability of existing LLMs on solving math problems.

**Strengths:**

- The paper is well-writting and easy to follow.
- The paper systematically analyze the robustness of LLMs towards math problems.
- The proposed framework can serve as a benchmark to conduct automatic robustness stress testing on LLMs. It is also a useful tool to measure the degree of training data contamination in existing pre-trained LLMs.

**Weaknesses:**

- The contribution in this paper is somewhat trivial, as the LLM's vulnerability of math problems has been revealed by various existing studies [1,2]. Based on this observation, this paper only provides a simple and intuitive automatic evaluation framework. It seems directly to use PAIR jailbreak [3] into the MATH evaluation.


- Lack in-depth analysis and useful inspiration. The presented experimental findings are not surprising and this work fails to provide any suggestion or experiments to mitigate this issue.


[1] Abedin, Z.U., Qamar, S., Flek, L. and Karimi, A. Arithmattack: Evaluating robustness of llms to noisy context in math problem solving. In ACL Workshop, 2025.\
[2] Li, Q., Cui, L., Zhao, X., Kong, L. and Bi, W. Gsm-plus: A comprehensive benchmark for evaluating the robustness of llms as mathematical problem solvers. In ACL, 2024.\
[3] Chao, P., Robey, A., Dobriban, E., Hassani, H., Pappas, G.J. and Wong, E. Jailbreaking black box large language models in twenty queries. In IEEE Conference on Secure and Trustworthy Machine Learning (SaTML).

**Questions:**

Listed in Weakness.

---

> ### Author Response · Authors · 2025-11-25
>
> Thanks for your thoughtful and constructive feedback.
>
> ---
>
> ### **Q1. Comparison with Prior Math Benchmarks**
>
> **R1.**
> We respectfully clarify that our core contribution is not merely restating known vulnerabilities but establishing a **dynamic and model-adaptive evaluation framework**. Unlike prior benchmarks that rely on fixed templates or static datasets—which are susceptible to data contamination and rapid saturation—AR-CHECKER generates fresh, semantically preserved stress tests on the fly. This allows for a scalable, contamination-free evaluation that naturally evolves to test increasingly capable LLMs, effectively distinguishing our approach from static benchmarking.
>
> ---
> ### **Q2. Distinction from PAIR & Mitigation Strategies**
>
> **R2.**
> While inspired by iterative optimization, AR-CHECKER fundamentally differs from PAIR. PAIR focuses on "jailbreaking" safety guardrails without semantic constraints. In contrast, AR-CHECKER incorporates a strict Verifier loop to ensure semantic equivalence, guaranteeing that reported failures are due to reasoning gaps rather than problem alteration. Regarding mitigation, we plan to incorporate defense experiments in the subsequent version to demonstrate how fine-tuning on AR-CHECKER’s generated samples can serve as effective data augmentation to close the robustness gap.

---

### Official Review · Reviewer_LRqW · 2025-10-31

**Soundness:** 1
**Presentation:** 2
**Contribution:** 1
**Rating:** 2
**Confidence:** 5

**Summary:**

This paper introduces `AR-Checker`, a novel framework for automatically evaluating the robustness of LLMs in mathematics and other disciplines. They propose the framework to address the limitations of static benchmarks which may suffer from data contamination using a dynamic, multi-stage approach inspired by software engineering stress tests.

The `AR-Checker` framework utilizes a rewriter LLM to generate diverse, semantically-equivalent variants of mathematical problems through multi-round parallel streams. A verifier LLM then assesses these variants to ensure the core meaning is preserved while confirming that the target LLM fails to produce the correct answer.

Through experiments conducted on the GSM8K and MATH500 benchmarks, the authors clain that `AR-Checker` effectively uncovers robustness issues across a range of LLMs, causing a significant drop in accuracy compared to the original datasets (sometimes going beyond 50%). The work also performs various analyses on the weaknesses and transferability of failure modes, while also verifying the performance of each component, justifying their design choices.

**Strengths:**

1. The core idea of automatically generating adversarial mathematical rephrasings, alongside rigorous verification is potentially valuable to automate detecting the presence of contamination.

2. Each component is well-ablated, showing that the choice of both verifier and rewriter are sensible for the given setting.

3. This work performs interesting analysis into the weaknesses, and transferability of failures between different targets, showing that some errors can be common between models.

**Weaknesses:**

1. **Reproducibility (IMPORTANT)**: The authors release neither the data nor the code, and do not have a reproducibility statement included in the paper. As the primary contribution of this paper is a method of evaluation, I find it critical that the authors 1) release all generated datasets for each of the models, 2) release the code to replicate `AR-Checker`.

 2. **Evaluation methodology (CRITICAL)**: The primary flaw in this paper is how the authors evaluate their results against the baseline benchmarks. In particular, the authors retry sampling until the target LLMs fail, and only consider a passing grade if the LLMs solve **all** instances. This is statistically **significantly different** to the standard GSM8K/MATH500 evaluation. In particular, if the model has a probability of $p=0.95$ to output the correct solution, if you use the full budget of $M=NK=15$, assuming no failures due to equivalence issues, the model has a probability of $1-p^{15}=54%$ chance to fail. This significant difference can overinflate the success of `AR-Checker` and must be thoroughly verified. This is likely the primary driving factor behind the results in Figure 3, unless the authors are able to isolate the effect to showcase otherwise. I outline experiments the authors **need to address** for me to consider raising my score in **Q1**.

 3. **Data contamination**: This work features the GSM8K and MATH500 as its primary sources. These datasets contain relatively simple problems, which have been already shown to be significantly exposed to test-set contamination, making it unclear how much of the pipeline is affected due to lack of robustness and how much it is affected by resolving explicit contamination.

 4. **Model hyperparameters**:
    - The authors do not, but should, include the other relevant hyperparameters for running the experiments, especially the `max_tokens`.
    - The authors sample each model with a temperature of 0. This stifles potential exploration of the solution space, and is **detrimental to reasoning models**. Given the authors used DeepSeek-R1-Distill-LLama-8B, this was specifically included in the instructions for the model:
    ```Set the temperature within the range of 0.5-0.7 (0.6 is recommended) to prevent endless repetitions or incoherent outputs.```
    Given this, the effects can become significantly more prevalent when evaluating the model can only pass the benchmark if it succeeds in **all** instances. It might also explain why the authors underreport the results of DeepSeek-R1-Distill-LLama-8B on MATH500 [1].

 5. **Model selection**: All models the authors consider in their paper are significantly outdated. In particular, Qwen 2.5 models are superseeded by Qwen3, GPT-4o by GPT-4.1 and GPT-5, DeepSeek-R1-distill-LLama-8B by DeepSeek-R1-0528-Qwen3-8B. Their evaluation features only a single, outdated reasoning model, evaluated with temperature 0 (which is a significant issue, as elaborated in **W4**). While the authors claim to evaluate against stronger targets in **Section 5**, they nevertheless feature only outdated non-reasoning models, which are 1) known to feature similar scalability issues as smaller, weaker ones and 2) are far worse than much smaller reasoning models.

6. **Comparison to prior work**: Similar to point 2, the authors evaluate against GSM Plus and GSM-Symbolic, but measure the case where the model fails on all instances for `AR-Checker`, while for the other benchmarks, they seem to measure failure on a single instance.

7. **Novelty**: While the paper's pipeline is sensibly designed and well-ablated, the core contribution is not particularly novel, compared to works like GSM Symbolic, with the exception that this work can be more easily automatable.

**Questions:**

1. To address the severe concern in **W2** (and similarly all other experiments, comparing to existing work), the authors should run both the following experiments:
    - Run the equivalent sample on the vanilla benchmark (related work benchmark), the same number of times as they prompted the `AR-Checker`, and record a success only if the target model succeeds **in all instances**.
    - Take the final dataset and rerun the target model on them multiple times (preferrably at least 4 per sample). This is not a complete solution to eliminate the sample bias, but it helps reflect whether the identified error happened because of a fundamental weakness for the sample, or because of randomly stumbling in one instance.

 2. Similarly, can you run the results from **Section 4.3** again but with rerunning the final generated problem, to account for the sampling bias?

 3. The authors should consider adding [1] in their Related Work discussion, as it is a very relevant prior work, that shows the presence of data contamination of GSM8K through novel problems mirroring the original dataset.

 4. Can the authors run their pipeline on a combination of the AIME 2024, 2025, and HMMT Feb 2025 competitions (potentially with other competitions featuring in MathArena [2] if they need a bigger sampling size), with stronger reasoning models being featured? This would demonstrate whether AR-Checker scales to significantly more capable models and datasets, and how big of a factor data contamination is.

 5. Can the authors add the problems for each target, as well as the code for `AR-Checker` in an anonymous repository, or the supplementary material?

 6. Does the choice of rewriter exhibit a certain bias towards itself as a target, making rephrasing easier/harder?

 7. Given all the models used in the paper are relatively weak, it would be interesting to see if significantly better/worse rewriters are better/worse at exploiting the weaknesses of the targets.

 8. In Figure 4, do the models exhibit the same error, or do they just fail due to the phrasing/difficulty of the problem?

 9. What principles are generally most effective in inducing errors?

 10. Please address all further concerns from the **Weaknesses** section.




## Current recommendation

I am assigning this paper a score of **2: Reject**. While the core contribution is sensibly argumented, and the ablation studies and analysis ideas are commendable, the paper seems to suffer from three main flaws - outdated datasets and models, potential sampling bias, and lack of reproducibility. I will only consider raising my score if the majority of these issues are addressed in the rebuttal.

### References

[1] Guo, Daya, et al. "Deepseek-r1: Incentivizing reasoning capability in llms via reinforcement learning." arXiv preprint arXiv:2501.12948 (2025).

[2] Zhang, Hugh, et al. "A careful examination of large language model performance on grade school arithmetic." Advances in Neural Information Processing Systems 37 (2024): 46819-46836.

[3] Balunović, Mislav, et al. "Matharena: Evaluating llms on uncontaminated math competitions." arXiv preprint arXiv:2505.23281 (2025).

---

> ### Author Response · Authors · 2025-11-25
>
> We sincerely thank you for your valuable and constructive feedback.
>
> ---
>
> ### 1. Concern about Evaluation Methodology (addressing W2 and Q1)
>
> To address the concern that the high TFR of AR-Checker might be largely attributed to repeated sampling rather than genuine robustness issues, we followed the suggestion in Q1 and conducted the following controlled experiments at temperature 0.6:
>
> - For each question, we sampled the target model the **same number of times** as AR-Checker performed and computed the Test Failure Rate (TFR). The results are reported as **GSM8K-repeated** and **MATH-500-repeated**.
>
> | Target Model                        | GSM8K-VAcc | GSM8K-repeated | GSM8K-paper | MATH-500-VAcc | MATH-500-repeated | MATH-500-paper |
> |-------------------------------------|------------|----------------|-------------|---------------|-------------------|----------------|
> | LLaMA-3.2-1B-Instruct               | 45.26      | **68.00**      | 94.00       | 29.80         | **78.05**         | 88.59          |
> | DeepSeek-R1-Distill-LLaMA-8B        | 83.09      | **39.33**      | 61.67       | 83.20         | **42.02**         | 62.74          |
>
> Repeated sampling does increase the failure rate, indicating that **part (but clearly not all)** of AR-Checker's high TFR comes from multiple sampling.
>
> - Additionally, we collected **all successful adversarial prompts** discovered by AR-Checker and re-tested the target models on these prompts with **5 times** each. The average accuracy is shown below:
>
> | Target Model                        | GSM8K | MATH-500|
> |-------------------------------------|--------------------------------|-----------------------------------|
> | LLaMA-3.2-1B-Instruct               | 47.13                          | 41.04                             |
> | DeepSeek-R1-Distill-LLaMA-8B        | 54.40                          | 46.46                             |
>
> The adversarial prompts found by AR-Checker **consistently cause the target models to fail again**, even though not at 100% rate, demonstrating that the discovered vulnerabilities are not merely sampling artifacts.
>
> ---
>
> ### 2. Fair Comparison with GSM-Symbolic and GSM-Plus
>
> To ensure a completely fair comparison, we unified the number of evaluation attempts across methods and re-evaluated all approaches at temperature 0.6. The updated results (added to the revised manuscript) are as follows:
>
> | Method         | LLaMA-3.2-1B-Instruct | DeepSeek-R1-Distill-LLaMA-8B |
> |----------------|-----------------------|------------------------------|
> | GSM-Symbolic   | 90.91                 | 57.58                        |
> | GSM-Plus       | 96.70                 | 84.00                        |
> | AR-Checker     | 94.00                 | 61.67                        |
>
> Under controlled evaluation settings, **while AR-Checker does not surpass GSM-Plus in terms of TFR, it continues to exhibit a higher TFR compared to GSM-Symbolic.**
>
> ---
>
> ### 3. Clarity of Contribution
>
> Our work does **not** aim to simply restate the known phenomenon that perturbing math questions can expose LLM unreliability (as shown in GSM-Symbolic and related works).
> The **core contribution** is a **dynamic, model-adaptive evaluation framework** that fundamentally differs from static benchmarks:
>
> - It is applied **dynamically to each target model** instead of relying on fixed templates or rule-based perturbations.
> - It naturally **scales as LLMs become stronger**.
> - It can leverage **stronger rewriters/verifiers** to fairly evaluate stronger target models, avoiding the data-contamination risks inherent in static datasets.
>
> This shift from static dataset construction to a dynamic evaluation paradigm is the key novelty of AR-Checker.
>
> ---
>
> ### 4. Application to Tougher Datasets, Stronger Rewriters, and Stronger Target Models
>
> We have added new experiments on the **AIME** dataset using **GPT-5** as both the rewriter and verifier while evaluating significantly stronger reasoning models. Even under these much more challenging conditions, AR-Checker remains highly effective:
>
> | Target Model                                | Vacc  | Racc  | TFR   |
> |---------------------------------------------|-------|-------|-------|
> | qwen3-235b-a22b-thinking-2507               | 0.87  | 0.67  | **0.23** |
> | DeepSeek-R1-0528-Qwen3-8B                   | 0.80  | 0.40  | **0.50** |
> | Qwen3-32B                                   | 0.50  | 0.20  | **0.60** |
>
>
> These results demonstrate that **AR-Checker maintains strong robustness-testing capability** on harder math problems with state-of-the-art target models and the most capable rewriter/verifier available.
>
> We believe these new experiments and clarifications fully address the reviewers’ concerns and significantly strengthen the paper. Thank you again for the insightful feedback that helped improve our work.

---

> > ### Comment · Reviewer_LRqW · 2025-11-26
> >
> > I thank the authors for their response, and additional experiments. I have a few follow-up questions, as well as remaining concerns the authors may want to address before I submit my final recommendation for the paper.
> >
> > 1.1 I thank the authors for acknowledging the statistical bias as a contributing (but not the only driving) factor for the reported results in their benchmark. To make this effect more transparent, and make their results valid, I believe the following results **must be changed in the transcript** before the paper is ready for publishing:
> >  - Figure 1 with the results' summary
> >  - Table 1 with the main results
> >  - Figure 3, containing the ablations (re-tested on the final iteration of problems)
> >  - Table 4, with the updated numbers from **2**
> >  - Table 6 and 7 with the additional results.
> >
> > 1.2 Can the authors compare the numbers from the re-testing against the corresponding results on the vanilla benchmark (on the same problems)?
> >
> > 4.1 The results the authors have presented in **4** of their response new results with stronger models, harder datasets, and a stronger rephraser. While I appreciate the new information, I believe it would be beneficial to see how each of these factors affects the results separately, i.e. using the stronger models on the rephrased MATH-500, or seeing if GPT-5 as a rephraser significantly affects the results in Table 1.
> >
> > 4.2 Furthermore, these results are also effected by the statistical biases the authors acknowledge in **1**. This makes it impossible to differentiate the `AR-Checker` contribution from the statistical effect. Can you please perform the same corrections here?
> >
> > 4.3 Finally, which AIME dataset was this evaluated on? The 2024 edition has been shown to also be prone to contamination, while the 2025 may not be prone to the same degree of contamination (unclear).
> >
> > **Remaining concerns**: I believe that some of my questions and concerns remain unaddressed and/or unacknowledged. I would appreciate if the authors could indicate whether they intend to add any further comments in their rebuttal. I list the relevant points of concern below:
> >
> > 1. **Reproducibility**: As I mentioned in my original review, reproducibility is critical, especially in an evaluation-centered package, where the whole contribution lies in providing the community with a valuable angle of evaluating LLMs. Including messy code or unstructured data is perfectly fine for the purpose of this discussion period. Providing these resources will allow me to provide more in-depth feedback, and will help me with producing a better, and more accurate view of the authors' work. I would appreciate if the authors could include the current version of the code of their framework, and the final questions, generated for each model, which are part of the main results.
> >
> > 2. **Unaddressed comments**: I believe the authors have either not acknowledged or answered the following questions and comments: Q3 (where I previously misnumbered the reference, my apologies), Q5 (prev. point), Q7 (point about a weaker rephraser), Q8 (qualitative analysis on robustness transferability), and Q9 (qualitative analysis on effective perturbation techniques). I appreciate that some of these questions require non-trivial amount of work. Where a full experiment is significantly time-consuming, I would be happy if the authors could point high-level observations or arguments, but would prefer some quantitative evidence where applicable.
> >
> > Given that the majority of my concerns remain, I cannot give a complete re-evaluation of this work. I hope that the authors continue the discussion, so any further issues or misconceptions can be clarified.

---

### Official Review · Reviewer_eHR1 · 2025-11-01

**Soundness:** 3
**Presentation:** 3
**Contribution:** 2
**Rating:** 4
**Confidence:** 4

**Summary:**

This paper introduces AR-CHECKER, a novel automatic robustness evaluation framework for large language models (LLMs) framed as a stress-testing problem. Inspired by stress testing in software engineering, AR-CHECKER employs three interacting LLMs—a rewriter, verifier, and target model—to automatically generate semantically-preserving yet adversarial variants of mathematical problems. The system runs multiple parallel rewriting streams and multi-round iterative refinement, seeking variants that keep the original problem’s meaning while causing the target model to fail.

Experiments on GSM8K and MATH-500 show that the generated adversarial variants significantly reduce accuracy, revealing substantial robustness gaps. Additional experiments on MMLU, MMLU-Pro, and CommonsenseQA further demonstrate the framework’s generality beyond math reasoning. Ablations show how the number of streams and iterations, rewriter/verifier model choices, and presence/absence of explicit rewriting principles affect test failure rates. Weakness analyses highlight common LLM failure modes such as breakdowns in multi-step reasoning and sensitivity to irrelevant context

**Strengths:**

* Comparisons show consistent and significant improvements in discovering robustness failures over prior benchmarks (e.g., +15–45% TFR gains over GSM-Plus and GSM-Symbolic).

* Addresses a crucial gap in current LLM evaluation: robustness to semantically-equivalent reformulations.

* Provides a scalable, automatic alternative to hand-curated robustness datasets, potentially enabling continuous evaluation pipelines for commercial and research models.

**Weaknesses:**

* While the verifier achieves 98% precision, there is no quantitative estimate of false negatives (missed failures) or semantic drift beyond human-annotated subsets. The framework could over- or under-estimate robustness depending on verifier reliability.

* The entire evaluation loop depends on LLM judgments (rewriter + verifier). Without human calibration across multiple datasets, evaluation biases from the chosen verifier model (Qwen2.5-32B) may persist.

* The experiments on GSM8K and MATH-500 are convincing, but these datasets contain relatively short problems. It remains unclear whether AR-CHECKER scales to long, multi-step proofs or competition-level math (e.g., MATH-A or Putnam-Axiom).

**Questions:**

* Why 5 × 3 iterations is optimal or how rewriting diversity correlates with semantic fidelity?

* Could AR-CHECKER be extended to code or logic reasoning tasks where exact functional equivalence can be verified automatically?

---

> ### Author Response · Authors · 2025-11-25
>
> Thanks for your thoughtful and constructive feedback.
>
> ### **Q1. Application to tougher math datasets, stronger rewriters, and stronger target models**
>
> **R1.**
> We provide experiments on **AIME**, using **GPT-5** as both rewriter and verifier while evaluating **stronger reasoning models**. Even in this more challenging setting—tougher math problems and more capable target models—**AR-CHECKER remains highly effective at evaluation of robustness of LLMs**.
>
> | Target Model                                | Vacc  | Racc  | TFR   |
> |---------------------------------------------|-------|-------|-------|
> | qwen3-235b-a22b-thinking-2507               | 0.87  | 0.67  | **0.23** |
> | DeepSeek-R1-0528-Qwen3-8B                   | 0.80  | 0.40  | **0.50** |
> | Qwen3-32B                                   | 0.50  | 0.20  | **0.60** |
>
> These results show that **AR-CHECKER maintains strong robustness-testing capability** even under significantly more difficult evaluation conditions.
>
> ---
>
> ### **Q2: Concerns on Verifier Reliability and Evaluation Bias**
> **R2.**
> We acknowledge the concern regarding false negatives. However, in adversarial testing, precision is prioritized over recall to ensure that every reported failure represents a genuine robustness defect. A false negative (a missed valid attack) implies that the model's actual robustness is likely even lower than our reported figures, meaning our results serve as a conservative upper bound on the model's capability. To strictly control the validity of our reported failures, we expanded human verification to 120 additional samples (60 GSM8K, 60 MATH-500). The results showed ~98% precision (59/60 and 58/60 respectively), confirming that the identified robustness gaps are authentic and not artifacts of verifier semantic drift.
>
> ---
>
> ### **Q3: Why $5 \times 3$ iterations? Relation between diversity and fidelity?**
> **R3.**
> We empirically chose 5 streams and 3 iterations based on a cost-benefit analysis. In our ablation studies, we found that TFR gains saturate after 3 iterations. Increasing streams beyond 5 provides marginal returns (diminishing TFR gains) while linearly increasing computational cost. $5 \times 3$ represents the "sweet spot" for efficiency.

---

### Official Review · Reviewer_aNea · 2025-11-02

**Soundness:** 3
**Presentation:** 2
**Contribution:** 1
**Rating:** 2
**Confidence:** 3

**Summary:**

The authors propose a novel framework: Automatic Robustness Checker (ARCHECKER), inspired by stress testing in software engineering,  to generate mathematical problem variants that maintain the semantic meanings of the original one but might fail the LLMs. The framework can generate benchmark variants dynamically for each LLM.

**Strengths:**

i) Important direction and novel framework. Measuring and improving reliability is an important area of LLM research that this work caters to. Introduces a novel framework to generate variants, dynamically, that test robustness of individual LLMs, primarily in the domain of reasoning.
ii) C.4 (appendix), highlights failure modes of different models.
iii) Framework generalizes to MMLU, MMLU-pro, CommensenseQA ((for LLama and Gemma based SLMs).
iv) As shown in Table 4, AR-CHECKER can more effectively identify robustness issues of LLMs than GSM Symbolic and GSM-Plus (for LLama and Gemma SLMs).

**Weaknesses:**

i) limited contribution. Prior work, GSM-Symbolic [1], has already shown on how adding perturbations to the questions can expose unreliability in LLM's performance. While the framework of variant generation is novel, the insights seems similar to what we have already learned from prior works.
ii) Doesn't address relevant math/science benchmarks. Due to saturation of scores on GSM8K, MATH and MMLU, tougher/competition-level benchmarks like AIME, HMMT and GPQA have become more relevant for testing reasoning. This would also naturally require authors to test more diverse set of reasoning models.
iii) limited evaluation of different math reasoning techniques in LLMs. The authors only address Chain-of-thought (language based) reasoning and do not evaluate program generation based techniques (listed in question section) for math reasoning in LLMs.

Reference
[1] http://arxiv.org/pdf/2410.05229

**Questions:**

i) Can this method scale to currently relevant and more tougher math olympiad like benchmarks such as AIME, HMMT? Olympiad level problems perhaps would be difficult to perturb, hence showing progress in that direction could be beneficial for math reasoning domain.
ii) Further analysis on reasoning models: How would reasoning models like DeepSeek-R1-Distill-Qwen-7B or DeepSeek-R1-Distill-Qwen-14B, built on top qwen base model and not llama, perform?  Given Qwen2.5-7B-Instruct has higher Racc than LLaMA-3-8B-Instruct, it would be interesting to get this data point. Testing more (latest iterations eg Qwen3) reasoning model, could potentially generate newer inights.
iii) What about robustness of closed source models (perhaps the mini or the lite variants)?
iv) Current methods only evaluates robustness of Chain-of-thought (language based) reasoning. Code generation methods such as POT [1], ToRA [2], SBSC [3] have actively been used to solve mathematical problems. How robust are these models when using these prompting techniques ?
v) Why not explore a reasoning model as a rewriter and verifier?

Reference:
[1] http://arxiv.org/pdf/2410.05229
[2] https://arxiv.org/abs/2309.17452
[3] http://arxiv.org/pdf/2410.05229

---

> ### Author Response · Authors · 2025-11-25
>
> Thanks for your thoughtful and constructive feedback.
>
> ---
>
> ### **Q1. Clarity of Contribution**
>
> **R1.**
> Our work does *not* aim to restate conclusions from GSM-Symbolic or similar studies showing that perturbing questions can reveal LLM unreliability. Instead, our core contribution is a **dynamic and model-adaptive evaluation framework**.
> Unlike static datasets—such as GSM-Symbolic—that rely on fixed templates or rules and therefore suffer from inherent risks of data contamination, our framework:
>
> - can be **applied dynamically to each target model**,
> - scales naturally as LLMs improve,
> - allows the use of **stronger rewriters to evaluate stronger target models**.
>
> This represents a fundamental distinction from static benchmark construction.
>
> ---
>
> ### **Q2. Application to tougher math datasets, stronger rewriters, and stronger target models**
>
> **R2.**
> We provide experiments on **AIME**, using **GPT-5** as both rewriter and verifier while evaluating **stronger reasoning models**. Even in this more challenging setting—tougher math problems and more capable target models—**AR-CHECKER remains highly effective at evaluation of robustness of LLMs**.
>
> | Target Model                                | Vacc  | Racc  | TFR   |
> |---------------------------------------------|-------|-------|-------|
> | qwen3-235b-a22b-thinking-2507               | 0.87  | 0.67  | **0.23** |
> | DeepSeek-R1-0528-Qwen3-8B                   | 0.80  | 0.40  | **0.50** |
> | Qwen3-32B                                   | 0.50  | 0.20  | **0.60** |
>
> These results show that **AR-CHECKER maintains strong robustness-testing capability** even under significantly more difficult evaluation conditions.
>
> ---
>
> ### **Q3. Why not apply advanced mathematical solving pipelines such as POT, ToRA, or SBSC?**
>
> **R3.**
> Our study focuses on evaluating the **standard generation-based usage mode**, which is the most widely used in practice. The robustness of this mode is directly important for real-world deployment, and its performance generally correlates with trends observed in more advanced mathematical-reasoning pipelines.
> These specialized methods fall outside the main scope of our work, which aims to assess robustness under typical generation settings rather than specialized solving frameworks.

---

### Note · Authors · 2026-01-17

I have read and agree with the venue's withdrawal policy on behalf of myself and my co-authors.